# Beta-Lactam Dose Optimisation in the Intensive Care Unit: Targets, Therapeutic Drug Monitoring and Toxicity

**DOI:** 10.3390/antibiotics12050870

**Published:** 2023-05-08

**Authors:** Amy Legg, Sinead Carmichael, Ming G. Chai, Jason A. Roberts, Menino O. Cotta

**Affiliations:** 1Menzies School of Health Research, Tiwi, Darwin, NT 0810, Australia; 2Herston Infectious Diseases Institute, Herston, Brisbane, QLD 4029, Australia; 3Royal Brisbane and Women’s Hospital, Departments of Intensive Care Medicine and Pharmacy, Brisbane, QLD 4029, Australia; 4Faculty of Medicine, University of Queensland Centre for Clinical Research (UQCCR), Brisbane, QLD 4029, Australia; 5Division of Anaesthesiology Critical Care Emergency and Pain Medicine, Nîmes University Hospital, University of Montpellier, 30029 Nîmes, France

**Keywords:** beta-lactam drugs, therapeutic drug monitoring, sepsis, intensive care unit, pharmacokinetics and pharmacodynamics

## Abstract

Beta-lactams are an important family of antibiotics used to treat infections and are commonly used in critically ill patients. Optimal use of these drugs in the intensive care unit (ICU) is important because of the serious complications from sepsis. Target beta-lactam antibiotic exposures may be chosen using fundamental principles of beta-lactam activity derived from pre-clinical and clinical studies, although the debate regarding optimal beta-lactam exposure targets is ongoing. Attainment of target exposures in the ICU requires overcoming significant pharmacokinetic (PK) and pharmacodynamic (PD) challenges. For beta-lactam drugs, the use of therapeutic drug monitoring (TDM) to confirm if the desired exposure targets are achieved has shown promise, but further data are required to determine if improvement in infection-related outcomes can be achieved. Additionally, beta-lactam TDM may be useful where a relationship exists between supratherapeutic antibiotic exposure and drug adverse effects. An ideal beta-lactam TDM service should endeavor to efficiently sample and report results in identified at-risk patients in a timely manner. Consensus beta-lactam PK/PD targets associated with optimal patient outcomes are lacking and should be a focus for future research.

## 1. Introduction

### 1.1. Bacterial Kill Characteristics of the Beta-Lactams

Beta-lactam antibiotics include penicillins, cephalosporins and carbapenems. These antibiotics are effective against a wide range of Gram-positive, Gram-negative and anaerobic bacteria. For the beta-lactam class of antibiotics, bacterial killing relates to the length of time that the drug concentration exceeds the minimum inhibitory concentration (MIC) of the causative organism at the site of infection. This principle was first described more than 70 years ago using pre-clinical data, where maximal efficacy occurred when drug concentrations were 2 to 5 times above the MIC [1,2,3]. In 1947, Dr Harry Eagle noted a plateauing of therapeutic effect once penicillin concentrations reached 10 times above the MIC for *Streptococcus pyogenes*, writing “Higher levels at the focus of infection represent largely waste penicillin, and lower levels have little if any therapeutic effect” [4]. Similarly, in experiments using *Staphylococcus aureus* isolates, no additional benefit was seen by increasing drug concentrations to 10 times above the MIC [5]. In these early experiments, a post-antibiotic effect was detected for Gram-positive organisms exposed to penicillins and cephalosporins, but not for Gram-negative organisms [6,7,8,9]. Subtherapeutic exposures of beta-lactam antibiotics were associated with the regrowth of resistant bacteria; for *Klebsiella* spp. exposed to cefepime, ceftazidime and meropenem, resistance prevention was achieved when trough concentrations were 1 to 4 times above the MIC [10].

Many of the pre-clinical studies determining pharmacodynamic (PD) targets for beta-lactams have been performed in neutropenic mouse thigh infection models [11,12,13,14,15,16,17,18]. Neutropenic models are used because the removal of the animal’s immune system ensures that bacterial efficacy can be more accurately linked to antibiotic exposure. However, this may result in higher PD targets than required in a host with a functional immune system [19]. Animal models can define pharmacokinetic (PK)/PD indices for bacteriostasis, 1-log_10_ bacterial kill and 2- or 3-log_10_ bacterial kill. The appropriate target for a specific pathogen, site of infection or timepoint in the course of infection (i.e., at the point of oral switch) has not been determined.

### 1.2. The Current Debate on Beta-Lactam PK/PD Targets in Critically Ill Patients

The accepted PK/PD index for beta-lactam antibiotics is the percentage of time the unbound drug is above the MIC (% fT > MIC) (i.e., ‘time dependent’). However, the optimal beta-lactam PK/PD target to achieve clinical cure and microbiological eradication in critically unwell patients remains undefined, including for different infusion durations.

In some studies, beta-lactam antibiotic exposure has been associated with clinical success and suppression of beta-lactam resistance when unbound plasma concentrations are maintained four to six times above the MIC throughout the entire dosing interval (100% fT > 4–6 × MIC) [20,21,22]. Other preclinical data have shown that the maximal bactericidal effect can be obtained with concentrations above the MIC for only a proportion of the dosing interval [13]. Using this theory, for intermittent dosing, an exposure target of beta-lactam unbound plasma concentration above the MIC for the entire dosing interval (100% fT > MIC) has been advocated to ensure that 40% to 70% fT > 4 × MIC is achieved [23]. However, whether 100% fT > MIC ensures 40% to 70% fT > 4 × MIC has been challenged, with a study using existing patient data to simulate first dose exposures suggesting that only a small number of critically ill patients will achieve 40–70% fT > 4 × MIC (piperacillin, ceftazidime and cefepime) when dosing to a 100% fT > MIC target [24].

An alternative exposure target of unbound concentrations four times above the MIC throughout the entire dosing interval (100% fT > 4 × MIC) for intermittent infusion, and a steady-state concentration of at least four times greater than the MIC (ƒCss ≥ 4 × MIC) for continuous infusion, was suggested to ensure maximal bacterial killing, prevent bacterial regrowth and ensure positive clinical outcomes [24]. The reality is that achieving those concentrations (i.e., an unbound cefepime trough concentration above 32 mg/L for *Pseudomonas* spp.) is problematic and risks drug toxicity. A single PK/PD target for each drug may be unrealistic, and, in fact, a series of targets may be needed, based on specific patient, pathogen, infection and drug administration factors.

### 1.3. Challenges among Critically Ill Patients

#### 1.3.1. Pharmacokinetic (PK) Alterations in the ICU

In critically ill patients, changes in physiologic parameters can result in significant drug PK derangement. Changes in volume of distribution due to endothelial damage, large volume fluid resuscitation, use of vasopressor drugs, decreased plasma protein concentrations (e.g., hypoalbuminemia) and changes in drug clearance due to alterations in kidney function affect hydrophilic antibiotics such as the beta-lactams [25]. A major risk factor for subtherapeutic drug exposure among critically ill patients relates to augmented renal clearance, generally defined as creatinine clearance (CrCl) of more than 130 mL/min/1.73 m^2^ [26].

The result of these PK alterations is significant inter- and intra-patient variability in drug concentrations within a population where achievement of adequate drug exposure is of utmost importance.

#### 1.3.2. Pharmacodynamic (PD) Challenges in the ICU

Rates of resistant pathogens are generally higher in the ICU compared to general hospital wards, related to the use of broad-spectrum antibiotics, transmission within the ICU and patients requiring invasive procedures [27]. Resistant pathogens represent a PD challenge, with elevated MICs requiring higher antibiotic concentrations to achieve the equivalent PK/PD target. For carbapenems, MICs have been shown to be several-fold higher for pathogens in the ICU compared to other hospital areas [28]. In the ICU, infections with resistant pathogens (e.g., beta-lactam-resistant *Klebsiella* spp. or carbapenem-resistant *Acinetobacter* spp.) are associated with a higher risk of mortality compared to other organisms [29].

#### 1.3.3. The Need for Beta-Lactam Dose Optimisation

Mortality from sepsis and septic shock remains high. A systemic review of articles from 2009 and 2019 in North America, Europe and Australia showed a 30-day mortality of 34.7% for patients with septic shock and 24.4% mortality from sepsis [30]. Given the negative clinical outcomes related to sepsis and septic shock, and the high PK variability among critically ill patients, an individualised dosing approach with therapeutic drug monitoring (TDM) should be considered [31].

## 2. Overview of Evidence for Usefulness of Beta-Lactam TDM in the ICU

Observational studies in the ICU have found that many patients needed dose adjustments to achieve 100% fT > MIC or 100% fT > 4 × MIC, but failure to achieve predetermined PK/PD targets is not always associated with negative clinical outcomes [32,33,34,35,36,37,38,39,40,41,42,43]. Despite their discordant findings, clinical observational studies provide useful information about the implementation of TDM. The patient groups most likely to benefit from TDM are those with either augmented renal clearance or kidney impairment or with modified Acute Physiology and Chronic Health Evaluation (mAPACHE) II scores between 9 and 22 [35,44,45,46,47,48,49,50,51]. Using actual MICs may be preferred to clinical breakpoint MICs, as actual MICs can be much lower than breakpoint MICs or epidemiological cut-off values (ECOFFs) [49]. An alternative approach is to use the MIC value plus two dilutions to avoid procedural variation between institutions, biological variation and test inaccuracies [52]. Certain beta-lactam antibiotics may be better candidates for TDM compared to others. Low target attainment has been seen for imipenem, piperacillin, ceftazidime and cefepime, as compared to meropenem [53,54,55,56]. As seen in a *post hoc* analysis of the Merino trial, isolates had piperacillin–tazobactam MICs much closer to the breakpoint than meropenem, so the impact of TDM may be more pronounced for piperacillin [57].

There have been a number of randomised controlled trials (RCTs) seeking to evaluate the impact of beta-lactam TDM in the ICU. De Waele et al. undertook a prospective, partially blinded RCT in Belgium between 2011 and 2012 [58]. Patients were randomised to either conventional therapy or the intervention, where daily TDM was used to adjust dosing. Piperacillin and meropenem were administered as extended infusions over 3 h. The PK/PD target was either 100% fT > 4 × MIC or 100% fT > MIC in the first 72 h of therapy. Forty-one patients were included, and significantly more patients in the intervention arm achieved the PK/PD targets at day 3 compared to the control group. None of the outcome parameters including clinical failure, bacterial persistence, change in SOFA score or hospital and 28-day mortality were statistically different between the two arms of the RCT.

Following this, Fournier et al. published results from their RCT conducted between 2013 and 2016, including 38 patients from a burns ICU [59]. Patients were randomised to either standard-of-care or intervention, where real-time TDM was performed, aiming for 100% fT > MIC (either from the actual organism or using MIC_90_ according to the European Committee on Antimicrobial Susceptibility Testing [EUCAST] Database). Trough concentrations within the target range were significantly higher in the intervention arm, with real-time TDM more than doubling the odds of meeting PK/PD targets; albeit without benefit in infection-related outcomes. Rates of clinical success were very high in both groups (92% intervention group vs. 97% control group) and MICs of identified pathogens were low (mean piperacillin MIC 2 mg/L and mean meropenem MIC 0.125 mg/L).

More recently, the TARGET trial was conducted between 2017 and 2019 in patients with severe sepsis and septic shock [60]. After enrolment and administration of a loading dose, patients were started on continuous infusion piperacillin–tazobactam and randomised to TDM (intervention) or no-TDM (control) arms. The target was 100% fT > 4 × MIC (if therapy was empiric, the MIC was assumed from the EUCAST epidemiological cut-off value [ECOFF] for *Pseudomonas aeruginosa*). The primary outcome was assessed using the mean daily sequential organ failure assessment (SOFA) score. In total, 249 patients were included (124 in the control group, 125 in the TDM group). Attainment of PK/PD targets improved with TDM (although still quite low, 37% vs. 15%), without significant benefit in SOFA scores, mortality or clinical or microbiological cure. Despite the improvement in target attainment, no difference in median piperacillin concentrations was seen. Mortality was linked to piperacillin concentrations; patients with the highest mortality had piperacillin concentrations over 96 mg/L, with a 28-day mortality rate over 4-fold higher than those with a piperacillin concentration between 32 and 64 mg/L (odds ratio 4.21, 95% CI 1.4–12.5; *p* = 0.01). This may be reflective of worse outcomes in patients with severe sepsis and subsequent kidney impairment.

The DOLPHIN trial was undertaken in eight sites across the Netherlands between 2018 and 2021 and included 388 patients prescribed either beta-lactam antibiotics or ciprofloxacin [61]. Patients were randomised to either standard dosing or intervention dosing, using TDM together with dosing software. The pharmacodynamic target for beta-lactams was 100% fT > MIC, with the upper limit being a trough concentration more than 10 times the MIC (100% fT > 10 × MIC). The MIC value was determined by using the EUCAST ECOFF breakpoint for the likely pathogen. The primary outcome of the DOLPHIN study was the length of stay (LOS) in the ICU, which was found to be similar between both groups. Unfortunately, rates of target attainment in the DOLPHIN trial were not statistically different between the standard dosing and TDM groups, making it difficult to measure a difference in outcomes related to differential drug exposure.

As yet, none of the RCTs have shown a difference in patient outcomes associated with TDM; however, failure to improve target attainment rates, use of surrogate MICs instead of actual MICs, broad inclusion criteria, small patient numbers and delay in TDM are significant limitations that will need to be addressed in future RCTs.

## 3. Excessive Exposure and Evidence of What Constitutes ‘Beta-Lactam-Associated Toxicity’

Although largely considered an antibiotic class with a wide therapeutic index, beta-lactams can cause a variety of exposure-related adverse drug reactions, including cytopenia, neurotoxicity, hepatotoxicity and nephrotoxicity. There is emerging evidence that some commonly used beta-lactam antibiotics may have concentration thresholds that are associated with an increased risk of toxicity (see Table 1). Idiosyncratic reactions have not been linked to drug exposure, so TDM is unlikely to be useful in this setting. Understanding where drug exposure–toxicity relationships exist is useful to guide the application of TDM when supratherapeutic concentrations is likely (i.e., in the setting of renal impairment) and drug toxicity is suspected. Evidence supporting drug exposure–toxicity relationships is limited to retrospective cohort studies and subject to confounding, including the assumption of drug causality.

### 3.1. Cytopenia

Benzylpenicillin dose and duration of treatment have been associated with neutropenia, potentially via immunologic or direct toxicity to cells, though the exact mechanism is unknown [62]. High doses (above 12 g per day) for longer than 2 weeks and low baseline neutrophil count have been associated with the development of neutropenia [63,64,65]. Haemolytic anaemia may also occur with benzylpenicillin. Traditionally, this has been associated with very high doses (over 20 MU [12 g] per day), although it has also been described with a standard dose of 8 MU (4.8 g) per day [62,66]. An association between dose and neutropenia suggests an exposure-related toxicity relationship may be present even though a concentration threshold for toxicity has not been established.

Duration of therapy has also been noted as a risk factor for the development of neutropenia with other penicillins [67]. One explanation for this finding is that beta-lactam degradation products contribute to the development of cytopenia, and these degradation products accumulate over time in infusion solutions [68,69,70]. Currently, TDM is not useful to predict neutropenia related to beta-lactam antibiotics; however, monitoring full blood count is suggested for those on high doses for a prolonged duration (>14 days) or with low baseline neutrophils.

### 3.2. Neurotoxicity

Neurotoxicity with penicillins may be related to their inhibitory effect on the gamma-aminobutyric acid (GABA) receptor [71]. Brain tissue interstitial fluid concentrations, rather than CSF concentrations, may predict neurotoxicity [72]. A retrospective review of dosing of cloxacillin and oxacillin in the ICU showed that 11/62 (18%) patients experienced neurotoxicity (delirium and persistent coma), and the range of trough concentrations in this group varied between 97 and 302 mg/L, much higher than efficacy target concentrations of 20–50 mg/L [73]. In a mixed cohort of ward and ICU patients, a total flucloxacillin trough concentration of 125.1 mg/L was associated with a 50% increased risk of developing neurotoxicity [74]. Flucloxacillin concentrations more than 10 times above the MIC were associated with neurotoxicity in a prospective cohort study (48% of patients were critically unwell), with an odds ratio of 1.12 for a 1 mg/L increase in mean flucloxacillin concentration (*p* = 0.02) [75].

In a retrospective study of 199 ICU patients, Beumier et al. found an increased incidence of neurotoxicity with median piperacillin trough concentrations exceeding 64 mg/L [76]. In another retrospective study of 53 ICU patients receiving continuous infusion of piperacillin–tazobactam (12 g/24 h), patients who developed neurotoxicity were more likely to have significant kidney impairment and higher total piperacillin concentrations at steady state compared to those who did not (156.9 mg/L vs. 91.3 mg/L, respectively, *p* = 0.0016) [77]. A receiver operating curve (ROC) analysis revealed a steady state concentration of >157.2 mg/L as a predictor of neurotoxicity. Imani et al. found higher total piperacillin trough concentrations in patients who developed neurotoxicity (150 mg/L vs. 75 mg/L, *p* < 0.01) as opposed to those who did not [74]. Logistic regression analysis identified a 50% risk of developing neurotoxicity when piperacillin total concentrations exceed 361.4 mg/L.

A potential relationship between cefepime exposures and neurotoxicity development in critically ill patients has been previously documented [78,79,80,81]. The variability in the definition of neurotoxicity, study population and the statistical methods used to determine toxicity thresholds makes it difficult to define an exact cefepime concentration threshold for toxicity. Collectively, the risk of toxicity appears to increase when trough cefepime concentrations exceed 20–30 mg/L [78,79,80,82,83,84]. If administered as a continuous infusion, steady state concentrations above either 35 mg/L or 60 mg/L have been associated with neurotoxicity [79,81].

For continuous infusion of cefazolin for treat osteomyelitis, toxicity was absent when total concentrations were below 100 mg/L, with only a single patient developing confusion with a cefazolin concentration of 127 mg/L [85]. Excessively high cefazolin doses given to patients with kidney impairment have resulted in seizure activity [86,87]. As described by Barretto et al. [88], although a specific cefazolin concentration is not associated with neurotoxicity, its chemical structure reveals a tetrazole moiety in the R1 position, as seen in pentylenetetrazole (a known neurotoxin). As such, patients should be monitored for neurotoxicity with cefazolin, though the role of TDM in this setting is unclear.

An increasing incidence of neurotoxicity was reported when median meropenem trough concentrations exceeded 16 mg/L [76]. Similarly, Imani et al. examined the effect of meropenem exposures on neurotoxicity; those who developed neurotoxicity had higher median trough concentrations and the probability of developing neurotoxicity was 50% when meropenem trough concentrations exceeded 64.2 mg/L [74]. Patients with cirrhosis had significantly higher piperacillin and meropenem concentrations and were at risk of worsening neurological status [89].

### 3.3. Hepatoxicity

Amoxicillin + clavulanate is frequently implicated as a cause of drug-induced liver injury [90]. This adverse reaction is likely related to a genetic predisposition associated with the HLA-DRB1*15 allele and may be protected by the HLA-DRB1*07 family of alleles [91]. Flucloxacillin has also been associated with hepatotoxicity, usually described as an immunogenic idiosyncratic adverse drug reaction associated with the presence of the HLA-B*57:01 allele [92]. One review of reports of hepatotoxicity to the Swedish Adverse Drug Reactions Advisory Committee described hepatoxicity with doses greater than 1.5 g/day of flucloxacillin [93]. Contrary to this finding, various publications have noted beta-lactam concentrations were not associated with hepatotoxicity [32,75,94,95]. Currently, as no clear concentration toxicity relationship exists for beta-lactams and hepatoxicity, TDM is unlikely to be helpful in this setting.

### 3.4. Nephrotoxicity

Penicillin antibiotics are considered a rare cause of iatrogenic nephrotoxicity, generally thought to be mediated by acute interstitial nephritis (AIN) from hypersensitivity, not dose [88,96]. Another proposed mechanism is drug accumulation in the proximal tubule causing direct toxicity [97]. Outside of the ICU, cloxacillin nephrotoxicity has been associated with plasma concentrations above 50 mg/L, though flucloxacillin concentrations have not been associated with nephrotoxicity [74,75,98].

Imani et al. reported higher trough concentrations in patients who developed nephrotoxicity as opposed to those who did not for piperacillin (130 mg/L vs. 65 mg/L, [*p* < 0.01]) and meropenem (25 mg/L vs. 10 mg/L, [*p* < 0.01]) [74]. Logistic regression analysis identified a 50% risk of developing nephrotoxicity when piperacillin total concentrations exceed 452.7 mg/L and total meropenem trough concentrations are above 44.45 mg/L. Of course, for beta-lactam antibiotics, higher concentrations would be expected in patients with impaired kidney function and are not necessarily causative. Additionally, as piperacillin and tazobactam can inhibit OAT1 and OAT3, increases in creatinine may not necessarily represent true kidney injury [99], weakening any potential association between piperacillin + tazobactam concentrations and nephrotoxicity.

### 3.5. Other Adverse Drug Reactions

Beta-lactam antibiotics can also cause allergy, thrombocytopenia and *Clostridioides difficile*-associated colitis, but an exposure–response relationship has not been described [62].

Although beta-lactam TDM is primarily considered for ensuring adequate concentrations and drug effectiveness, monitoring for exposure-related toxicity is increasingly being recognised as a valuable role of TDM. For patients with organ impairment, or for beta-lactam antibiotics with a documented exposure–toxicity relationship (e.g., cefepime and neurotoxicity), regular clinical assessment and use of TDM should be considered to minimise drug toxicity.

**Table 1 antibiotics-12-00870-t001:** Maximum concentrations for commonly used for beta-lactam drugs.

Drug	Maximum Unbound Trough Concentration	Toxicity with Reported Associated with Supratherapeutic Concentration
Flucloxacillin	20 mg/L	neurotoxicity [75] *
Cefepime	20 mg/L	neurotoxicity [100]
Piperacillin	130 mg/L	neurotoxicity, nephrotoxicity [74]
Meropenem	44 mg/L	neurotoxicity, nephrotoxicity [74]

* Based on expert opinion; concentrations above 20 mg/L are more than 10 times above the EUCAST ECOFF for oxacillin.

## 4. The Ideal Service for Beta-Lactam TDM in the ICU

An ideal beta-lactam TDM service in the ICU will involve a balance of resource management, appropriate patient selection, timeliness of sampling and reporting of results, along with expert interpretation and dose modification (see Figure 1). Ongoing quality assessment of each factor is suggested to ensure a service remains ideal for the local circumstances. Given that up to 70% of ICU patients are administered at least one antibiotic, patient selection is crucial [29]. Not all patients within the ICU are critically ill or maintain deranged physiology during their admission; therefore, TDM resources, if available, should be directed towards those most likely to benefit.

Access to real-time microbiology and MICs to direct TDM targets is ideal; however, patients often may not have an identified causative organism within 48 h of treatment initiation, and many laboratories do not routinely report MICs [101,102]. Furthermore, interpretation of an MIC value is not straightforward and should consider microbiology laboratory accuracy limitations [52]. Once patients and pathogens are identified, access to reliable laboratory services with quick turnaround times is necessary, preferably with same-day results to allow for review, interpretation and prompt dose modification, if required. While there are several high-performance liquid chromatography and liquid chromatography-mass spectrometry assays available, access to these in ICUs may be limited [31]. One option to negate the need for chromatography assays is point-of-care (POC) testing; aiming to provide rapid results at the bedside, allowing immediate clinical interpretation [103]. While most research into POC TDM has focused on immunosuppressants, recent studies with beta-lactam antibiotics show promise [104]. Measurement of unbound (i.e., pharmacologically active) beta-lactam antibiotic concentrations is required in ICU patients, especially for those beta-lactam antibiotics that are highly protein bound such as flucloxacillin, ceftriaxone and cefazolin [101,105].

TDM results require clinical interpretation. Clinician-led dose modification has been shown to increase the proportion of patients who reach PK/PD targets, albeit with significant inter-clinician variability [58,106]. Using dosing software can enhance the utility of TDM. Bayesian forecasting software shows promise in optimising dosing to improve efficacy and minimise toxicity, but its use is limited by cost of the software and the adequate training of clinicians in its use [105]. Access to validated dosing software coupled with trained clinicians to interpret TDM results and use software to guide dosing recommendations supports an ideal TDM service within the ICU. As with all clinical services, the costs associated with implementing and maintaining a TDM service need to be weighed up with the potential benefits. These costs are not limited to just analytical costs, but also include collecting, storing and transporting samples, adequate resourcing with skilled professionals to interpret results and potentially dosing software systems to facilitate dose adjustment and model-informed precision dosing.

**Figure 1 antibiotics-12-00870-f001:**
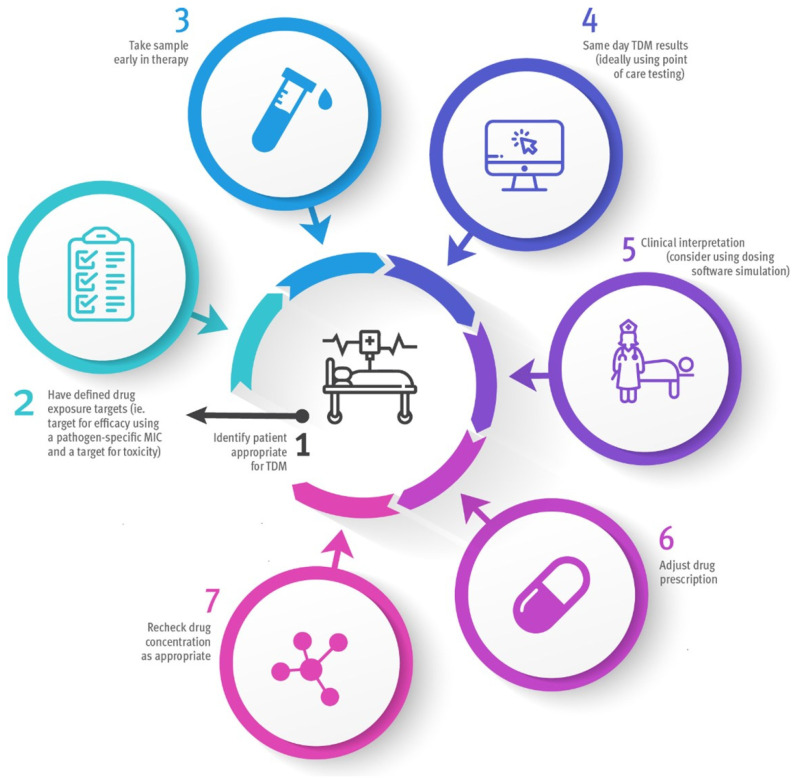
Ideal approach to TDM in the ICU. TDM: therapeutic drug monitoring; MIC: minimum inhibitory concentration.

## 5. Areas for Future Research

Given their widespread use in ICUs, optimising beta-lactam antibiotic dosing in critically ill patients remains important [107]. A robust analysis of the relationship between beta-lactam exposure and clinical outcomes is still lacking. This research should be prioritised so that beta-lactam PK/PD targets specific to patients with sepsis are accurately defined. Together with patient morbidity and mortality, outcomes such as clinical cure, microbiological eradication and development of antibiotic resistance require exploration in future studies, as well as cost-effectiveness analysis.

Globally, beta-lactam TDM is becoming more prominent in the ICU [108]. Research in streamlining processes that lead to improvements in turn-around times for reporting TDM results, however, is urgently required. Timely reporting of beta-lactam TDM results can then be used by ICU clinicians to expedite dose optimisation strategies. Further to this, optimisation strategies such as the use of dosing nomograms and/or model-informed precision dosing software require thorough investigation so that any patient- and system-level benefits associated with these strategies can be quantified.

In settings where TDM is unavailable, model-informed precision dosing using *a priori* prediction algorithms in dosing software packages may be a useful approach in the ICU [109]. However, limited data are available quantifying the clinical benefit of this approach.

## 6. Conclusions

More data are needed before the value of beta-lactam TDM in the ICU can be understood. Currently, its role consists of ensuring adequate concentrations for effectiveness (where the optimal target for this is unknown and varies between institutions) and checking that toxicity thresholds are not being exceeded. Coupled with the rapid turnaround of results, the use of TDM early in beta-lactam therapy may provide the greatest clinical benefit for critically ill patients.

Guidelines for TDM should identify patients most likely to benefit from TDM and report targets for clinical effectiveness and toxicity. Every TDM result requires clinical interpretation and action; improved accuracy and uptake of dosing software will be valuable when designing new dosing regimens.

Further research to robustly define beta-lactam PK/PD targets based on improvements in clinical outcomes is imperative to ensure that a meaningful assessment of TDM-based interventions can be undertaken.

## Data Availability

Not applicable.

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
