# Peer review of "Beta-Lactam Dose Optimisation in the Intensive Care Unit: Targets, Therapeutic Drug Monitoring and Toxicity"

_antibiotics, 2023, doi:10.3390/antibiotics12050870_

Round 1

Reviewer 1 Report

The authors present a well-written and thorough review of current data and importance of further research into the need for therapeutic drug monitoring in ICUs. Although current data does not suggest much of a clinical benefit to TDM for beta-lactams in the ICU, there are significant pitfalls to many of the studies in this area as discussed by the authors. The manuscript is thorough, but concise and provides an important overview of the subject.

Some comments below:

A brief discussion of what a beta-lactam and the spectrum of organisms they are effective against may be helpful in the introduction for some readers.

It may also be helpful to discuss relative costs of TDM given that there may not be a clinical benefit to the expensive monitoring process.

Title: I believe TDM and ICU should not be abbreviated in the title.

Page 4 second paragraph – enrollment is misspelt.

Overall, the manuscript would be of significant value to many readers and highlights an important topic In medicine. A better understanding of whether TDM with beta-lactams is cost-effective and clinically beneficial is of great concern to the healthcare system. I believe the manuscript is fit for publication with minor revisions and grammatical corrections.

Author Response

We appreciate the considered reviews from all reviewers and have addressed comments individually as detailed below:

Reviewer 1:

The authors present a well-written and thorough review of current data and importance of further research into the need for therapeutic drug monitoring in ICUs. Although current data does not suggest much of a clinical benefit to TDM for beta-lactams in the ICU, there are significant pitfalls to many of the studies in this area as discussed by the authors. The manuscript is thorough, but concise and provides an important overview of the subject.

Some comments below:

A brief discussion of what a beta-lactam and the spectrum of organisms they are effective against may be helpful in the introduction for some readers.

Authors’ response: Thank you for the suggestions. We have now added text in the Introduction describing the groups of antibiotics that constitute the beta-lactam class as well as their broad range of antibacterial activity. Beta-lactam antibiotics include penicillins, cephalosporins and carbapenems. These antibiotics are effective against a wide range of Gram-positive, Gram-negative and anaerobic bacteria.

It may also be helpful to discuss relative costs of TDM given that there may not be a clinical benefit to the expensive monitoring process.

Authors’ response: We appreciate this comment and have added information about cost (including less commonly considered costs in the ‘Ideal TDM service’ section: ‘As with all clinical services, the costs associated with implementing and maintaining a TDM service needs to be weighed against potential benefits. These costs are not limited to just analytical costs, but also includes collecting, storing and transporting samples, adequate resourcing with skilled professionals to interpret results, and potentially, dosing software systems to facilitate dose adjustment and model-informed precision dosing.’

Title: I believe TDM and ICU should not be abbreviated in the title

Authors’ response: We have now amended the title as per the Reviewer’s suggestion, with thanks. The title now reads as: Beta-lactam dose optimisation in the intensive care unit: targets, therapeutic drug monitoring and toxicity

Page 4 second paragraph – enrollment is misspelt –

Authors’ response: Thanks for comment – ‘enrolment’ is the British spelling, while ‘enrollment’ is the American version, so we will leave any potential amendment up to the discretion of the Antibiotics journal editorial team.

Overall, the manuscript would be of significant value to many readers and highlights an important topic in medicine. A better understanding of whether TDM with beta-lactams is cost-effective and clinically beneficial is of great concern to the healthcare system. I believe the manuscript is fit for publication with minor revisions and grammatical corrections.

Authors’ response: Many thanks for this positive feedback.

Reviewer 2 Report

Thank you for the opportunity to review this manuscript. This is a review of Beta-lactam dose optimisation and/or therapeutic drug monitoring in the intensive care setting.  In recent years, many reviews of therapeutic drug monitoring have been published, nearly all of which endorse therapeutic drug monitoring yet describe the uncertainty in defining the targets one should aim for and the lack of evidence of improved clinical outcomes

PMID: 36551496 

PMID: 36532752

PMID: 35370708

PMID: 34798631 

PMID: 34136822

PMID: 33480024 

PMID: 33463382

PMID: 32597263

PMID: 32537644 

PMID: 31585474

Only one  (of which I’m aware) takes a critical approach to the literature;  PMID: 35815181

Furthermore, a recent systematic review/meta-analysis of patient outcomes from studies of beta-lactam TDM also do not find evidence of improved patient outcome. PMID: 35731853,

and a large RCT (albeit not combined RCT of ciprofloxacin and beta-lactams) also do not show improved clinical outcomes.  (PMID: 36350354 ).  The former has now been updated with the later beta-lactam data and results are unsurprisingly not supportive of beta-lactam TDM ,   (Zeggil, T and Dalton, B,  Clinical Infectious Diseases, in press)

I have only checked citation accuracy for the discussions in this review in some selected areas (eg. Page 2 – PK/PD targets, specifics below) and found cited references to be inaccurate and/or irrelevant.

I found unsupported recommendations for targets to avoid toxicity.

The discussion on toxicity avoidance also includes irrelevant information about idiosyncratic reactions.

I can agree with ongoing research to determine IF TDM is useful.   But should we endorse a resource intensive procedure such as TDM outside the research setting in the absence of evidence that it improves patient outcomes?  Improved attainment of a poorly defined target only helps clinicians feel good about dosing precision.  We can likely agree if TDM were a drug it would not be approved with this level of evidence.  However  (if you’ll indulge a cranky old reviewer ;)), if TDM were a drug with no biologic activity other than it changed beta-lactam PK such that it always achieved targets, would we endorse it with evidence that we currently have regarding target achievement?  I would argue: No, absolutely not without high quality clinical data that demonstrates achievement of such targets really improves outcomes.  The only argument for TDM of beta-lactams (outside of research setting) is that it is unlikely to be harmful.  Yet wasting time, resources and distracting clinicians from evidence based activities is potentially harmful insidiously and in ways that are poorly defined and discredit the professions of pharmacy and pharmacology.

This review would be a value to the literature if there is critical synthesis of existing literature and /or if the “ideal TDM service” discussed on page 7-8 was supported by citations that had implemented such a service and proved benefits beyond increased target attainment.  

There is no doubt that a good deal of effort was put into this review.  However, unfortunately, due to the low scholastic level, lack of critical thinking and novel additional information (as detailed below), I do not feel this project is a worthy addition to existing literature, nor is mature enough to warrant publication.  I feel it would be best for the authors to start fresh, rather than trying to revise .

Detailed review;

In the abstract, there are several points that require greater detail  eg. “For each patient, the appropriate exposure (concentration) targets for efficacy and toxicity need to be decided.  “   I sympathize regarding the difficulties of communication within the word limits of the abstract, but, it should be stated if there are evidence based means of deciding individualized targets for efficacy and toxicity.  The authors are no doubt aware of the uncertainty in the literature on critical care population targets, I am unaware of means of setting individualized targets other than educated guess work based on patient gestalt.

After discussion in lines above about uncertainty of targets there is the sentence “An ideal beta-lactam TDM service should endeavour to efficiently sample and report results in identified at-risk patients in a timely manner. “  It is unclear if this is an endorsement of TDM services (outside of research settings) in critical care for beta-lactams. If so ( and based on reading the rest of the article, I think it is), is this not putting the cart before the horse as adjusting a drug dose to achieve a target would essentially be treating a number not the patient? If not, it should be explicitly stated that TDM of beta-lactams is not recommended outside of the research setting, and should not be prioritized ahead of more established clinical activities.

Page 2 third paragraph “Clinical targets for beta-lactam antibiotic exposure associated with microbiological eradication have been shown to correlate with unbound plasma concentrations main- tained 4 to 6 times above the MIC throughout the dosing interval (100% fT> 4-6 x MIC) [10, 20, 21]

This does not make any sense. Please re-word or delete the first part of sentence. Ie. “Microbiologic eradication has been associated with unbound plasma concentrations….”

The references cited do not seem to me to support the claim of a clinical target of 100% fT>4-6xMIC.  Does it not make more sense to discuss studies such as the DALI study of clinical response to levels of fT>MIC (PMID: 24429437 ) led by the senior author of this review than the references selected for the reasons discussed below?

Reference 10 studied Cmin/MIC ratios associated with resistance development in an in vitro model. This is important PD/PK work but does it relate to fT>MIC and microbiologic eradication?

Reference 20 studied various PD/PK parameters in modeling experiment of meropenem for adult LRTI . There were few patients who did not achieve 100% fT>MIC and per figure 2A responding  and non responding patients had the same range & distribution of values.     Figure concluded “fC(min)/MIC was the statistically significant parameter associated with meropenem clinical and microbiological response in the adults with LRTI. “

Reference 21 studied Cmin/MIC ratios of meropenem required to suppress resistant mutants in a in vitro model. This is important PD/PK work but does it relate to fT>MIC and microbiologic eradication?

…..However, whether 100% fT > MIC ensures 40% to 70% fT > 4 x MIC has been challenged, with a study using existing patient data to simulate first dose exposures suggesting that only a small number of criti- cally ill patients will achieve 40-70% fT > 4 x MIC (piperacillin, ceftazidime and cefepime) when dosing to a 100% fT > MIC target [23]. “  While it is broadly understood that early appropriate antibiotic therapy improves patient outcome,  what is the evidence that  first dose achievement of target has clinical importance ? Furthermore, how does one reconcile the lack of actual MIC at the time when (usually) first doses are given?  Usually in septic patients in ICU or on the way to ICU, first does are given immediately after blood cultures are taken. 

Page 4, paragraph 4, “As yet, none of the RCTs have shown a difference in patient outcomes associated with TDM, however, failure to improve target attainment rates, use of surrogate MICs instead of actual MICs, broad inclusion criteria, small patient numbers and delay in TDM are significant limitations that will need to be addressed in future RCTs. “ Agree more research is required to determine if TDM for beta-lactams in ICU is going to be helpful, so why endorse beta-lactam TDM in critical care outside of the research setting?

Page 4, paragraph 5, It would be valuable to readers if an introductory paragraph discussed the limitations of studies demonstrating establishing dose-toxicity relationships.  Are any prospective? How is causation established (if at all)? How are sepsis, metabolic derangement, psychoactive substances and other drugs ruled out as a potential cause of the adverse event being studied? For instance, the study of cefepime associated neurotoxicity by T Huwyler et al (PMID: 28111294), used the WHO-Uppsala Monitoring Centre system [11] to establish causality and adverse events for which cefepime was considered possible we analyzed as cefepime toxicity.  If one looks at the criteria of possible in the WHO-Uppsala monitoring centre system (https://cdn.who.int/media/docs/default-source/medicines/pharmacovigilance/whocausality-assessment.pdf?sfvrsn=5d8130bb_2&download=true);

Event or laboratory test abnormality, with reasonable time relationship to drug intake • Could also be explained by disease or other drugs • Information on drug withdrawal may be lacking or unclear”

then it is really unclear that cefepime was not simply an innocent bystander guilty by association.  In other studies cited here, how was causality established?

Drug toxicity is often considered acceptable risk in critically ill patients because of high mortality.  Are there cases when achievement of efficacy targets exceed the recommended toxicity? (eg. Piperacillin at 100%4xfT>MIC , if MIC is 8

It would be valuable for readers to discuss number needed to harm or absolute risk  in a toxic blood level range.

Page 4, Table 1.  Are these trough concentrations, random, peak? Could you provide dose administration method appropriate recommendations for each drug? (Intermittent vs continuous infusion)?  What is the source/citation of these recommendations? What population(s) were these levels established in? For readers it would be valuable if a column was inserted listing the toxicity to be avoided by maintenance of dosing below the threshold limit.

Page 5, first paragraph.  This paragraph is a review of evidence associating daily dose and duration.  “An association between dose and neutro- penia suggests an exposure-related toxicity relationship may be present even though a concentration threshold for toxicity has not been established. “  If there are known associations with duration and daily dose why would a blood level be needed?  Isn’t it easier, less resource intensive to determine the daily dose and duration?

Page 6, 4 th paragraph. Amox/clav and fluclox  hepatoxicity are associated with specific genotype and daily dosing (respectively),… or perhaps not. In any case there is no discussion of a blood level threshold associated with hepatotoxicity. Why would TDM be useful (outside of research)?  Suggest deletion of this paragraph.

Page 6, 5th paragraph. Nephrotoxicity.  As is pointed out concentrations may be higher because of nephrotoxicity not the other way around.  If penicillins are rare causes of nephrotoxicity, what is the value in TDM (NNH)?

Page 6, 6th paragraph. Other adverse drug reactions. … “monitoring for exposure-related toxicity is increasingly being recognised as a valuable role of TDM. “ Is there a citation(s) to support this or is it opinion of the authors?

If these ADR have not been associated with drug levels (let alone proven causative), what value does TDM have? 

Page 8, 3rd paragraph “Given their widespread use in ICUs, optimising beta-lactam antibiotic dosing in crit- ically ill patients remains important [104]. A robust analysis of the relationship between beta-lactam exposure and clinical outcomes is still lacking.

The second sentence contradicts the first sentence if “optimizing” involves aiming for a beta-lactam exposure target.

Page 8, 5th paragraph. “Currently, its role consists of ensuring adequate concentrations for effectiveness (where the optimal target for this is unknown and varies between institutions) and check- ing that toxicity thresholds are not being exceeded “ 

How can one ensure adequate concentrations if adequate concentrations are undefined?

Author Response

Reviewer 2:

Thank you for the opportunity to review this manuscript. This is a review of Beta-lactam dose optimisation and/or therapeutic drug monitoring in the intensive care setting. In recent years, many reviews of therapeutic drug monitoring have been published, nearly all of which endorse therapeutic drug monitoring yet describe the uncertainty in defining the targets one should aim for and the lack of evidence of improved clinical outcomes

PMID: 36551496 

PMID: 36532752

PMID: 35370708

PMID: 34798631 

PMID: 34136822

PMID: 33480024 

PMID: 33463382

PMID: 32597263

PMID: 32537644 

PMID: 31585474

Only one (of which I’m aware) takes a critical approach to the literature; PMID: 35815181

Furthermore, a recent systematic review/meta-analysis of patient outcomes from studies of beta-lactam TDM also do not find evidence of improved patient outcome. PMID: 35731853,

and a large RCT (albeit not combined RCT of ciprofloxacin and beta-lactams) also do not show improved clinical outcomes.  (PMID: 36350354).  The former has now been updated with the later beta-lactam data and results are unsurprisingly not supportive of beta-lactam TDM, (Zeggil, T and Dalton, B,Clinical Infectious Diseases, in press).

I have only checked citation accuracy for the discussions in this review in some selected areas (eg. Page 2 – PK/PD targets, specifics below) and found cited references to be inaccurate and/or irrelevant.

I found unsupported recommendations for targets to avoid toxicity.

The discussion on toxicity avoidance also includes irrelevant information about idiosyncratic reactions.

I can agree with ongoing research to determine IF TDM is useful.   But should we endorse a resource intensive procedure such as TDM outside the research setting in the absence of evidence that it improves patient outcomes?  Improved attainment of a poorly defined target only helps clinicians feel good about dosing precision.  We can likely agree if TDM were a drug it would not be approved with this level of evidence.  However, (if you’ll indulge a cranky old reviewer ;)), if TDM were a drug with no biologic activity other than it changed beta-lactam PK such that it always achieved targets, would we endorse it with evidence that we currently have regarding target achievement?  I would argue: No, absolutely not without high quality clinical data that demonstrates achievement of such targets really improves outcomes.  The only argument for TDM of beta-lactams (outside of research setting) is that it is unlikely to be harmful.  Yet wasting time, resources and distracting clinicians from evidence-based activities is potentially harmful insidiously and in ways that are poorly defined and discredit the professions of pharmacy and pharmacology.

This review would be a value to the literature if there is critical synthesis of existing literature and /or if the “ideal TDM service” discussed on page 7-8 was supported by citations that had implemented such a service and proved benefits beyond increased target attainment.  

There is no doubt that a good deal of effort was put into this review.  However, unfortunately, due to the low scholastic level, lack of critical thinking and novel additional information (as detailed below), I do not feel this project is a worthy addition to existing literature, nor is mature enough to warrant publication.  I feel it would be best for the authors to start fresh, rather than trying to revise.

Authors’ response: Before providing individual responses to the Reviewer, we would highlight the fact that the Reviewer’s comments are in stark contrast to the other two Reviewers. Furthermore, the Reviewer highlights 10 other papers that support our general themes before saying our paper had not been written critically? Does this also suggest that the other 10 separate papers that came to the conclusion were not sufficiently critical because they came to the same conclusion? This confused writing is common throughout the Reviewer’s comments which only serve to celebrate their own opinion – this is of course where a lack of critical thinking and introspection is very clear.

In our view, a retrospective review of risk factors (i.e. PMID: 35815181) without any pharmacokinetic data is an exceptionally low form of evidence. In no way does this contradict the findings of extensive literature reviews included in the position statements from international societies. 

AND

Without wanting to offend the authors of the recent letter in Clinical Infectious Diseases, the Reviewer’s highlighting of this recent meta-analysis as an 'updated' justification of his/her interpretation of the area is insufficient to us. We would point out that this is a letter, not a peer reviewed paper, that combines observational studies with underpowered RCTs and as such provides no new data that can help us understand the potential value of the intervention. The more relevant meta-analysis which the Reviewer has curiously elected not to cite has strong scientific quality in terms of peer review and a meta-analysis methodology focusing only on RCTs and has been published in Clinical Microbiology and Infection (Codina et al, CMI 2023 doi: 10.1016/j.cmi.2023.03.018). This paper concluded “This meta-analysis demonstrates that target attainment, treatment failure, and nephrotoxicity were significantly improved in patients who underwent individualised antimicrobial dose optimisation. However, it did not show a significant decrease in mortality, clinical cure or microbiological outcome.” Again, this very favourably supports our viewpoint and uses a much higher level of scientific rigour than that selectively used by the Reviewer.

Detailed review;

In the abstract, there are several points that require greater detail eg. “For each patient, the appropriate exposure (concentration) targets for efficacy and toxicity need to be decided.” I sympathize regarding the difficulties of communication within the word limits of the abstract, but, it should be stated if there are evidence based means of deciding individualized targets for efficacy and toxicity.  The authors are no doubt aware of the uncertainty in the literature on critical care population targets, I am unaware of means of setting individualized targets other than educated guess work based on patient gestalt.

Authors’ response: We have removed this sentence and updated the Abstract. Although debate is ongoing about targets, beta-lactam TDM in the ICU does occur and we believe our Review provides the clinician with key points should they choose to utilise this option.

After discussion in lines above about uncertainty of targets there is the sentence “An ideal beta-lactam TDM service should endeavour to efficiently sample and report results in identified at-risk patients in a timely manner. “  It is unclear if this is an endorsement of TDM services (outside of research settings) in critical care for beta-lactams. If so (and based on reading the rest of the article, I think it is), is this not putting the cart before the horse as adjusting a drug dose to achieve a target would essentially be treating a number not the patient? If not, it should be explicitly stated that TDM of beta-lactams is not recommended outside of the research setting, and should not be prioritized ahead of more established clinical activities.

Authors’ response: Our intention is to provide some guidance on what an ideal service for beta-lactam TDM may look like. As per our comment above, it is up to clinician discretion on whether they choose to use beta-lactam TDM.

Page 2 third paragraph “Clinical targets for beta-lactam antibiotic exposure associated with microbiological eradication have been shown to correlate with unbound plasma concentrations main- tained 4 to 6 times above the MIC throughout the dosing interval (100% fT> 4-6 x MIC) [10, 20, 21]

This does not make any sense. Please re-word or delete the first part of sentence. Ie. “Microbiologic eradication has been associated with unbound plasma concentrations….”

The references cited do not seem to me to support the claim of a clinical target of 100% fT>4-6xMIC.  Does it not make more sense to discuss studies such as the DALI study of clinical response to levels of fT>MIC (PMID: 24429437 ) led by the senior author of this review than the references selected for the reasons discussed below?

Reference 10 studied Cmin/MIC ratios associated with resistance development in an in vitro model. This is important PD/PK work but does it relate to fT>MIC and microbiologic eradication?

Reference 20 studied various PD/PK parameters in modeling experiment of meropenem for adult LRTI . There were few patients who did not achieve 100% fT>MIC and per figure 2A responding  and non responding patients had the same range & distribution of values.     Figure concluded “fC(min)/MIC was the statistically significant parameter associated with meropenem clinical and microbiological response in the adults with LRTI. “

Reference 21 studied Cmin/MIC ratios of meropenem required to suppress resistant mutants in a in vitro model. This is important PD/PK work but does it relate to fT>MIC and microbiologic eradication?

Authors’ response: We have now amended the sentence to “In some studies, beta-lactam antibiotic exposure has been associated with clinical success and suppression of beta-lactam resistance when unbound plasma concentrations are maintained four to six times above the MIC throughout the entire dosing interval (100% fT> 4-6 x MIC).” We have also updated the references, specifically addition of Reference 20 to the clinical study:

Tam, V.H., et al., Pharmacodynamics of cefepime in patients with Gram-negative infections. J Antimicrob Chemother, 2002. 50(3): p. 425-8.

…..However, whether 100% fT > MIC ensures 40% to 70% fT > 4 x MIC has been challenged, with a study using existing patient data to simulate first dose exposures suggesting that only a small number of criti- cally ill patients will achieve 40-70% fT > 4 x MIC (piperacillin, ceftazidime and cefepime) when dosing to a 100% fT > MIC target [23]. “  While it is broadly understood that early appropriate antibiotic therapy improves patient outcome,  what is the evidence that  first dose achievement of target has clinical importance ? Furthermore, how does one reconcile the lack of actual MIC at the time when (usually) first doses are given?  Usually in septic patients in ICU or on the way to ICU, first does are given immediately after blood cultures are taken. 

Authors’ response: There is no evidence that target achievement after the 1st dose is of clinical importance. There is, however, enough evidence to support early use of effective antibiotic therapy (including adequate dosing) to circumvent treatment failure.

The premise that the target of 100% fT > MIC is unlikely to achieve the exposure of 40% to 70% fT > 4 x MIC (derived from preclinical studies) remains, and thus continues the discussion of whether the theoretical target of 100% fT > MIC is applicable if one is to use it as a marker to achieve 40% to 70% fT > 4 x MIC.

Authors’ response: Regarding the lack of actual MIC available at the start of (empiric-based) beta-lactam therapy, we have touched upon this further in the manuscript whereby we emphasise the relevance of using actual MICs (if available) rather than clinical breakpoint MICs (that would be used in the absence of an isolated pathogen or as pointed out, at the beginning of empiric-based beta-lactam antibiotic therapy).

Page 4, paragraph 4, “As yet, none of the RCTs have shown a difference in patient outcomes associated with TDM, however, failure to improve target attainment rates, use of surrogate MICs instead of actual MICs, broad inclusion criteria, small patient numbers and delay in TDM are significant limitations that will need to be addressed in future RCTs. “Agree more research is required to determine if TDM for beta-lactams in ICU is going to be helpful, so why endorse beta-lactam TDM in critical care outside of the research setting?

Authors’ response: We have provided our response re: endorsement of beta-lactam TDM in our first two responses to Reviewer 2 and will avoid repeating here.

Page 4, paragraph 5, It would be valuable to readers if an introductory paragraph discussed the limitations of studies demonstrating establishing dose-toxicity relationships.  Are any prospective? How is causation established (if at all)? How are sepsis, metabolic derangement, psychoactive substances and other drugs ruled out as a potential cause of the adverse event being studied? For instance, the study of cefepime associated neurotoxicity by T Huwyler et al (PMID: 28111294), used the WHO-Uppsala Monitoring Centre system [11] to establish causality and adverse events for which cefepime was considered possible we analyzed as cefepime toxicity.  If one looks at the criteria of possible in the WHO-Uppsala monitoring centre system (https://cdn.who.int/media/docs/default-source/medicines/pharmacovigilance/whocausality-assessment.pdf?sfvrsn=5d8130bb_2&download=true);

“Event or laboratory test abnormality, with reasonable time relationship to drug intake • Could also be explained by disease or other drugs • Information on drug withdrawal may be lacking or unclear”

then it is really unclear that cefepime was not simply an innocent bystander guilty by association.  In other studies cited here, how was causality established?

Drug toxicity is often considered acceptable risk in critically ill patients because of high mortality.  Are there cases when achievement of efficacy targets exceed the recommended toxicity? (eg. Piperacillin at 100%4xfT>MIC , if MIC is 8

It would be valuable for readers to discuss number needed to harm or absolute risk in a toxic blood level range.

Authors’ response: While this level of information may be valuable, the current manuscript is an overview – these detailed considerations would require a publication with a different scope. We have added information about limitations of the data and highlighting the likely concentration dependant toxicities as suggested by the reviewer.

Page 4, Table 1.  Are these trough concentrations, random, peak? Could you provide dose administration method appropriate recommendations for each drug? (Intermittent vs continuous infusion)?  What is the source/citation of these recommendations? What population(s) were these levels established in? For readers it would be valuable if a column was inserted listing the toxicity to be avoided by maintenance of dosing below the threshold limit.

Authors’ response: We have amended the table headings with “trough concentrations” as well as removing two cephalosporins antibiotics for which current toxicity data is limited.

Page 5, first paragraph.  This paragraph is a review of evidence associating daily dose and duration.  “An association between dose and neutropenia suggests an exposure-related toxicity relationship may be present even though a concentration threshold for toxicity has not been established. “  If there are known associations with duration and daily dose why would a blood level be needed?  Isn’t it easier, less resource intensive to determine the daily dose and duration?

Authors’ response: This is valuable clinical information about drug dose and duration and risk of neutropenia, we have never suggested TDM be performed in the text. We have now explicitly stated this as perhaps we assumed too much baseline knowledge from the readership.

 Page 6, 4 th paragraph. Amox/clav and fluclox  hepatoxicity are associated with specific genotype and daily dosing (respectively),… or perhaps not. In any case there is no discussion of a blood level threshold associated with hepatotoxicity. Why would TDM be useful (outside of research)?  Suggest deletion of this paragraph.

Authors’ response: We believe it is useful to know when to abstain from using beta-lactam TDM and we thought that would be obvious in the setting of an idiosyncratic reaction. We have added a sentence at the end of this section for clarity.

Page 6, 5th paragraph. Nephrotoxicity.  As is pointed out concentrations may be higher because of nephrotoxicity not the other way around.  If penicillins are rare causes of nephrotoxicity, what is the value in TDM (NNH)?

Authors’ response: We have added a sentence at the end of this section for clarity. 

Page 6, 6th paragraph. Other adverse drug reactions. … “monitoring for exposure-related toxicity is increasingly being recognised as a valuable role of TDM. “ Is there a citation(s) to support this or is it opinion of the authors? If these ADR have not been associated with drug levels (let alone proven causative), what value does TDM have? 

Authors’ response: As noted above, we have added a statement in the first paragraph of this section noting that the role of TDM in drug induced toxicity is in the setting of a clear concentration-exposure toxicity relationship.

Page 8, 3rd paragraph “Given their widespread use in ICUs, optimising beta-lactam antibiotic dosing in critically ill patients remains important [104]. A robust analysis of the relationship between beta-lactam exposure and clinical outcomes is still lacking.

The second sentence contradicts the first sentence if “optimizing” involves aiming for a beta-lactam exposure target.

Authors’ response: We do not feel these statements are contradictory, given that, even in the absence of data for TDM, administration of an appropriate dose of an effective antimicrobial is part of standard of care.

Page 8, 5th paragraph. “Currently, its role consists of ensuring adequate concentrations for effectiveness (where the optimal target for this is unknown and varies between institutions) and checking that toxicity thresholds are not being exceeded “ 

How can one ensure adequate concentrations if adequate concentrations are undefined?

Authors’ response: This area of uncertainty appears in clinical practice in ICU frequently. In practice, we may be able to determine those patients in whom there is concern about subtherapeutic or supratherapeutic concentrations and decide on a reasonable PK-PD target based on clinical data and disease-related risk factors. We believe our review provides some guidance to help clinicians interpret this information. 

Reviewer 3 Report

My comments are as follows:

Comment 1: P.2 L6 There is a typographical error.

However, this may result in higher PD targets than required in a host with a functional immune system. However, this may result in higher PD targets than required in a host with a functional immune system [14, 19].

Comment 2:

Some hospitals cannot actually measure drug concentration. Even in such a hospital, we have to treat seriously ill patients. Instead of "real TDM" based on observed values, "virtual TDM" based on predicted values is also considered as an alternative. I think that the usefulness of TDM can be shown more by giving an example of "virtual TDM". Please consider it.

Author Response

Reviewer 3:

Comment 1: P.2 L6 There is a typographical error.

However, this may result in higher PD targets than required in a host with a functional immune system. However, this may result in higher PD targets than required in a host with a functional immune system [14, 19]

Authors’ response: Thank you for highlighting the typographical error. We have now corrected it.

Comment 2:

Some hospitals cannot actually measure drug concentration. Even in such a hospital, we have to treat seriously ill patients. Instead of "real TDM" based on observed values, "virtual TDM" based on predicted values is also considered as an alternative. I think that the usefulness of TDM can be shown more by giving an example of "virtual TDM". Please consider it.

Authors’ response: Thank you for your recommendation. We have now added a sentence at the end of ‘Areas for future research’: “In settings where TDM is unavailable, model-informed precision dosing using a priori prediction algorithms in dosing software packages may be a useful approach in the ICU [109]. However, limited data are available quantifying the clinical benefit of this approach.”